# *Plasmodium falciparum* sexual parasites regulate infected erythrocyte permeability

Guillaume Bouyer[1,2,7], Daniela Barbieri[2,3,7], Florian Dupuy[2,3], Anthony Marteau[2,3], Abdoulaye Sissoko [2,4], Marie-Esther N'Dri[2,3], Gaelle Neveu[2,3], Laurianne Bedault[2,3], Nabiha Khodabux[2,3], Diana Roman[2,4], Sandrine Houzé[2,4], Giulia Siciliano[5], Pietro Alano[5], Rafael M. Martins[6], Jose-Juan Lopez-Rubio [6], Jérome Clain[2,4], Romain Duval[2,4], Stéphane Egée[1,2,8] & Catherine Lavazec [2,3,8 ✉]

To ensure the transport of nutrients necessary for their survival, *Plasmodium falciparum* parasites increase erythrocyte permeability to diverse solutes. These new permeation pathways (NPPs) have been extensively characterized in the pathogenic asexual parasite stages, however the existence of NPPs has never been investigated in gametocytes, the sexual stages responsible for transmission to mosquitoes. Here, we show that NPPs are still active in erythrocytes infected with immature gametocytes and that this activity declines along gametocyte maturation. Our results indicate that NPPs are regulated by cyclic AMP (cAMP) signaling cascade, and that the decrease in cAMP levels in mature stages results in a slowdown of NPP activity. We also show that NPPs facilitate the uptake of artemisinin derivatives and that phosphodiesterase (PDE) inhibitors can reactivate NPPs and increase drug uptake in mature gametocytes. These processes are predicted to play a key role in *P. falciparum* gametocyte biology and susceptibility to antimalarials.

[1] Sorbonne Université, CNRS UMR 8227, Station Biologique de Roscoff, Roscoff, France. [2] Laboratoire d'excellence GR-Ex, Paris, France. [3] Université de Paris, Inserm U1016, CNRS UMR 8104, Institut Cochin, Paris, France. [4] Université de Paris, IRD 261, MERIT, Paris, France. [5] Istituto Superiore di Sanità, Roma, Italy. [6] Université de Montpellier 1 & 2, CNRS 5290, IRD 224, MIVEGEC, Montpellier, France. [7] These authors contributed equally: Guillaume Bouyer, Daniela Barbieri. [8] These authors jointly supervised this work: Stéphane Egée, Catherine Lavazec. ✉email: catherine.lavazec@inserm.fr

Malaria remains a major public health problem with more than 200 million cases and almost half a million deaths annually. The clinical symptoms of malaria are attributed to *Plasmodium* asexual stages, whereas parasite transmission from humans to mosquitoes relies on the gametocytes, the specialized sexual cells formed by a fraction of parasites which cease asexual propagation. *Plasmodium falciparum* gametocyte maturation requires about ten days, and is classically divided into five developmental stages based upon morphological features[1]. If eradication of malaria obviously requires compounds targeting asexual stages, transmission remains the Achilles heel of the strategies implemented today[2]. However, most antimalarial drugs target the asexual stages, but they are less effective against gametocytes[3,4]. As a consequence, infected individuals remain a source of transmission even after they are cured. It remains unclear why gametocytes become less sensitive to artemisinin as their maturation progresses from stage I to stage V. Thus, understanding the biology of gametocyte development within erythrocytes is crucial for successful malaria elimination.

Following *P. falciparum* invasion, the infected erythrocyte displays important alterations of its membrane properties. For instance, asexual parasites increase erythrocyte permeability to diverse solutes to ensure the transport of nutrients and waste products necessary for their replication and survival. Mature asexual parasites activate weakly selective anion channels in the erythrocyte membrane to generate new permeability pathways (NPPs) that render infected erythrocytes more permeable to a range of nutrients[5,6] and to several antimalarials[7–10]. Key proteins involved in NPPs have been characterized, like parasitic proteins CLAG3/RhopH1, RhopH2 and RhopH3 or endogenous constituent of the peripheral-type benzodiazepine receptor[11–14]. However, the full identity of the NPPs has never been conclusively established[15]. Although their regulatory mechanisms also remain unclear, cyclic AMP (cAMP)/Protein Kinase A (PKA) pathway seems to play a key part in regulating ion channels in the membrane of erythrocytes infected with asexual stages[16]. Despite the vital role of NPPs in asexual stages, the level of NPP activity in gametocyte-infected erythrocytes (GIE) is unknown, nor is the role of NPPs in antimalarials uptake by GIE. Refractoriness of GIE to lysis upon short exposure to isosmotic sorbitol solution has led to the dogma that NPPs are totally absent in gametocyte stages[17]. However, this assumption is not supported by the fact that during their 10-day maturation, gametocytes also need to absorb nutrients from the plasma and get rid of toxic waste products they generate upon hemoglobin digestion, two major roles of NPPs at asexual stages. In addition, the absence of NPPs in GIE is not consistent with the expression profile of members of the RhopH complex that are synthesized in mature intracellular parasites and then secreted upon egress onto the erythrocyte targeted for invasion[18,19]. Therefore, all known NPP components should be present at the surface of newly invaded GIE.

In this study, we performed isosmotic lysis, electrophysiology, fluorescence tracer uptake and viability experiments to evaluate NPP activity during *P. falciparum* gametocytogenesis. We found that NPP activity is regulated by the cycling AMP signaling cascade, and interfering with this pathway can reactivate erythrocyte permeability and facilitate uptake of artemisinin derivatives by mature gametocytes.

## Results

**NPPs are still active in immature gametocytes**. To evaluate the NPP activity in GIE, we first performed measurements of isosmotic lysis of immature GIE in sorbitol, a sugar alcohol permeant through NPPs[6]. Sorbitol uptake was drastically reduced in stage II GIE compared to trophozoites and schizonts; however, about 36% GIE were lysed after 60 min in sorbitol, suggesting that erythrocyte permeability is modified by immature gametocytes (Fig. 1a). Lysis kinetics in stage II GIE were similar to that of late rings 16 h-post-invasion, when NPPs start to be expressed during the asexual cycle[20]. GIE lysis was significantly inhibited by the general anion channel inhibitors 5-nitro-2-(3-phenylpropylamino)benzoic acid (NPPB) and furosemide, by the benzodiazepine Ro5-4864 and the isoquinoline PK11195, which have all been shown to inhibit NPPs (Fig. 1b)[11,21]. Equivalent lysis kinetics observed in alanine or phenyltrimethylammonium (PhTMA$^+$) isosmotic solutions, with a slightly greater lysis in PhTMA$^+$ than in sorbitol, confirmed that uptake occurred through NPPs (Fig. 1c). Measurement of membrane currents using whole-cell patch-clamp strengthened observations obtained with isosmotic lysis. Quantification of total ion fluxes across the host membrane of individual infected erythrocytes indicated that membrane conductance at $-100$ mV was sixfold lower in stage II GIE than in trophozoites (Fig. 1d and Supplementary Fig. S1). However, conductance in early GIE showed anion selectivity, inward rectification and NPPB sensitivity (Fig. 1e and Supplementary Fig. S1), which are three fundamental characteristics of the correlate membrane currents of NPP activity[22]. These observations are consistent with detection of the NPP component RhopH2[12] in immature GIE by immunostaining (Supplementary Fig. S1). Pharmacological inhibition of NPPs is expected to lead to nutrient starvation and accumulation of toxic metabolic wastes eventually leading to parasite death. Thus, we measured viability of early gametocytes (stages I−II) upon NPPs inhibition by using a luciferase-based assay with a transgenic parasite expressing luciferase under the control of the gametocyte-specific promoter *Pfs16*[23]. We observed a slight but significant decrease in early gametocytes viability 48 h after a 3-h exposure to 100 μM NPPB, consistent with a vital role of NPP activity (Fig. 1f). Fewer transition to later stages gametocytes after exposure to NPPB was also quantified by Giemsa staining (Supplementary Fig. S1). Therefore, organic solute uptake, patch-clamp and viability studies indicate that NPPs are weaker than in trophozoites but are still active and necessary in early GIE.

**NPPs decline along gametocytogenesis**. To further analyze the evolution of this permeability along gametocytogenesis, we performed isosmotic lysis experiments in sorbitol, alanine and PhTMA$^+$ for synchronous cultures of stage I, II, III, IV, and V gametocytes (Fig. 2a–c). A transgenic parasite expressing GFP under the control of the *Pfs16* promoter was used to discriminate stage I gametocytes from asexual stages[24]. From stage I to stage V, we observed a progressive decrease in permeability for the three different solutes, leading to undetectable NPP activity in mature GIE. Electrophysiological experiments using whole-cell configuration performed on GIE from all stages confirmed that membrane conductance drops along gametocyte maturation, leading to currents in mature GIE corresponding to levels usually recorded in uninfected erythrocytes (Fig. 2d and Supplementary Fig. S2). Altogether, these data indicate that NPP activity declines along gametocytogenesis.

**NPP activity is regulated by cAMP-signaling**. We then aimed to determine the mechanisms underlying the regulation of NPP activity in GIE. We have previously observed that cAMP/PKA pathway activates ion channels in the membrane of erythrocytes infected with asexual stages[16], suggesting that cAMP-signaling may also modulate NPPs during gametocytogenesis. To address this hypothesis, we performed sorbitol lysis experiments on early GIE preincubated with KT5720 and H89, two independent and

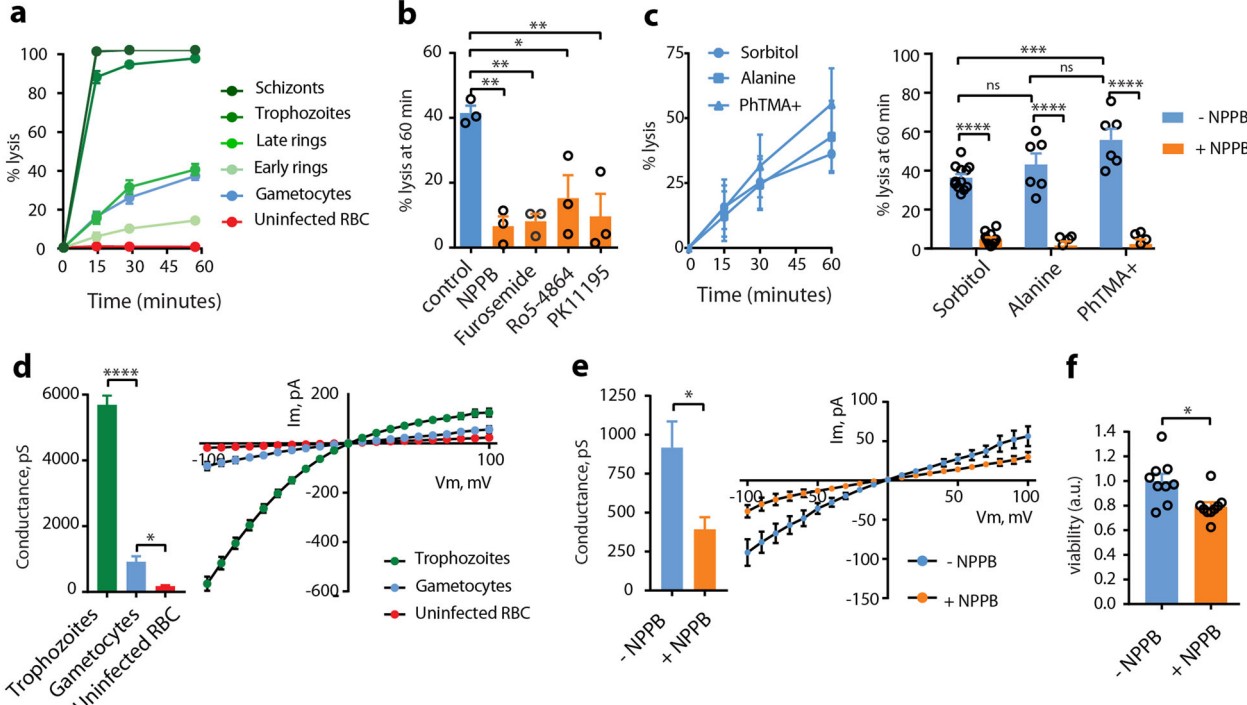

**Fig. 1 NPPs are still active in immature gametocytes. a** Kinetics of sorbitol-induced isosmotic lysis of erythrocytes infected with early rings (4 h-post-invasion (hpi)), late rings (16 hpi), trophozoites (28 hpi), schizonts (40 hpi), stage II gametocytes and uninfected erythrocytes during 60 min. $n = 3$ independent experiments. **b** % lysis of stage II GIE in sorbitol at 60 min with or without 100 μM NPPB, Furosemide, Ro5-4864, or PK11195. **c** Lysis kinetics (left) and % lysis at 60 min (right) of stage II GIE in sorbitol, alanine or PhTMA+. **d** Patch-clamp experiments on erythrocytes infected with trophozoites, (green bar, number of cells = 14), stage II gametocytes (blue bar, number of cells = 14) or uninfected erythrocytes (red bar, number of cells = 11). Left: whole-cell conductance calculated at −100 mV. Right: I−V plot from patch-clamp experiments. **e** Patch-clamp experiments on stage II GIE in presence or absence of NPPB (number of cells = 14 and 5, respectively). Left: whole-cell conductance calculated at −100 mV. Right: I−V plot from patch-clamp experiments. **f** Viability (luciferase activity) of early gametocytes of the NF54-cg6-pfs16-CBG99 line 48 h after a 3-h incubation with or without 100 μM NPPB. The graph shows relative viability normalized by the average luciferase activity of control (without NPPB). a.u. arbitrary units. In (**b**, **c**, **f**), circles indicate the number of independent experiments. In (**d**, **e**), the data distribution is shown in Supplementary Fig. S1. Error bars show the standard error of the mean (SEM). Statistical significance is determined by a Mann−Whitney test (**d**−**f**) or by one-way ANOVA with Dunnet correction (**b**) or Sidak correction (**c**) for multiple comparisons. ****$p < 0.0001$, ***$p < 0.001$, **$p < 0.01$, *$p < 0.05$, ns: non-significant difference.

widely used PKA inhibitors that have already been shown to inhibit PKA activity in *P. falciparum*[25,26]. We found that both compounds ablated GIE permeability (Fig. 3a), whereas the cAMP analog 8-Bromide-cyclic adenosine-monophosphate (8Br-cAMP) significantly increased the sorbitol-induced lysis (Fig. 3b). As an alternative way to investigate the effects of inhibiting PKA activity, we used a transgenic parasite that overexpresses the regulatory subunit of PKA using an episome selected by the antimalarial drug pyrimethamine (pHL*pfpkar*)[16,26]. The down-regulation of PKA activity in this parasite line also resulted in a significant decrease in permeability of early GIE (Fig. 3c). This phenotype was reverted to the levels of wild-type parasites upon incubation with 8Br-cAMP, or in a revertant parasite line that has shed the overexpressing episome[26] (Fig. 3c). We observed a similar inhibition of alanine- or PhTMA+-induced isosmotic lysis of stage II GIE for the pHL*pfpkar* line (Supplementary Fig. S3). These results indicate that PKA activity contributes to NPPs in immature GIE, as described for asexual stages[16].

These observations suggest that the decline in NPP activity during gametocytogenesis likely results from the previously described drop in cAMP concentration in GIE due to the rising expression of PfPDEδ in mature gametocytes[26]. Thus, we hypothesized that interfering with cAMP pathway with molecules that raise cAMP levels may turn on NPP activity in stage V GIE. As expected, raising cAMP levels in mature GIE upon incubation

with 8Br-cAMP or with the PDE inhibitors sildenafil and tadalafil restored the sorbitol-induced lysis to the levels observed in early GIE, whereas tadalafil did not trigger any significant effect in early GIE (Fig. 3d, e and Supplementary Fig. S4). Consistently, the loss of PDE activity in a transgenic parasite in which the *Pfpdeδ* gene had been deleted[27] also resulted in a drastic increase in permeability of mature GIE (Fig. 3f and Supplementary Fig. S3). These findings show that activating the cAMP/PKA pathway can reactivate NPPs in mature stages.

**NPPs contribute to the uptake of artemisinin derivatives by immature GIE.** *P. falciparum* mature gametocytes are known to become less sensitive to several antimalarials, including artemisinin derivatives, as their maturation progresses from stage I to stage V[28,29]. Although lower metabolic activity and complete hemoglobin digestion in mature stages have been suggested to contribute to this decrease in chemosensitivity[4,30], we hypothesized that this decrease may also be linked to the decline of host membrane permeability during gametocyte maturation. To address this hypothesis, we first evaluated whether NPP activity is required for uptake of artemisinin derivatives by immature GIE. For this purpose, we used Fluo-DHA, a fluorescent probe mimicking the clinical antimalarials dihydroartemisinin (DHA) and artemether[31]. This probe was synthesized from DHA and a 4-nitrobenzoxadiazole (NBD) fluorophore (Supplementary

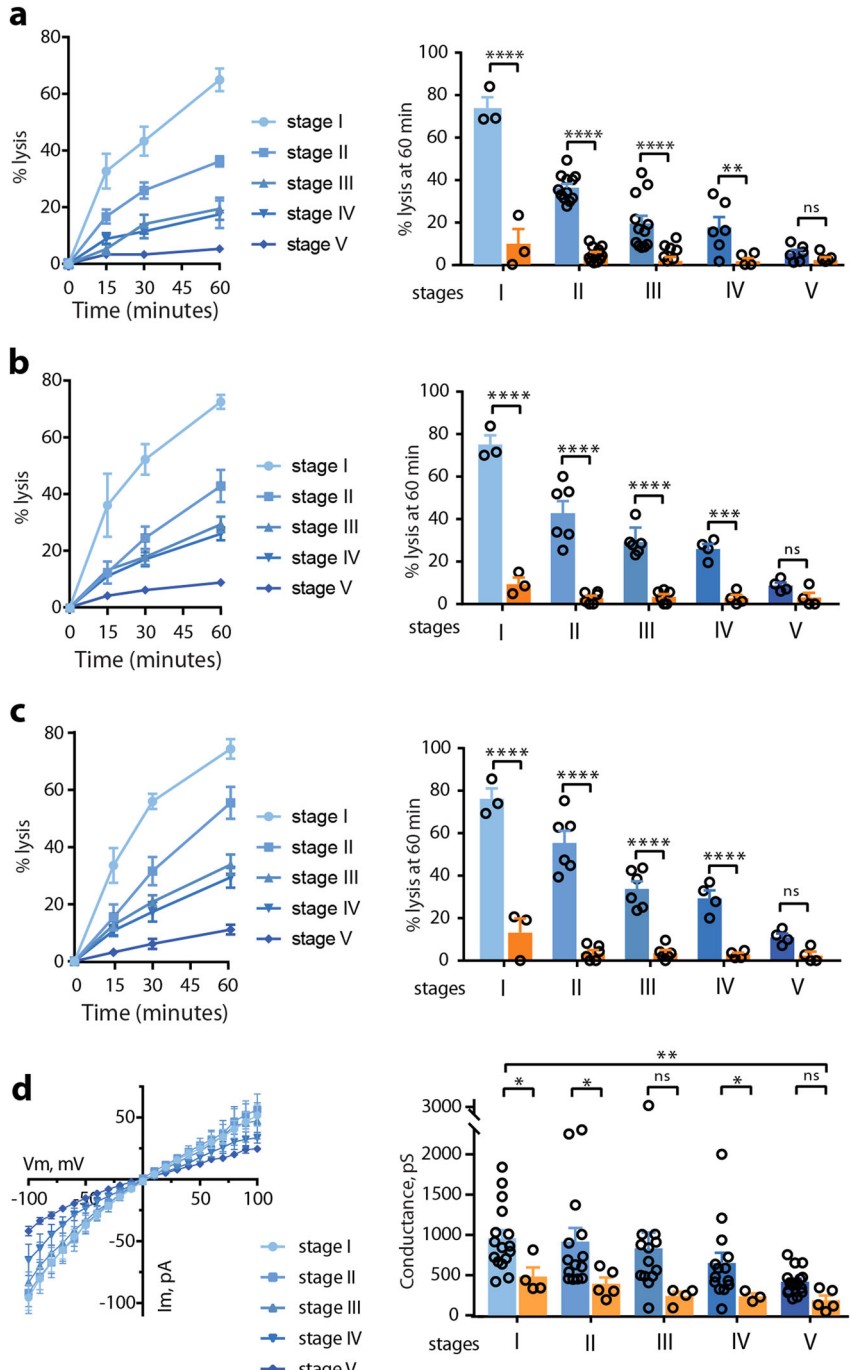

**Fig. 2 NPPs decline along gametocytogenesis. a–c** Sorbitol-induced (**a**), alanine-induced (**b**) and PhTMA$^+$-induced (**c**) isosmotic lysis of GIE from stage I to stage V. Left: Kinetics of isosmotic lysis during 60 min. Right: % lysis at 60 min with (orange) or without (blue) 100 μM NPPB. Circles indicate the number of independent experiments and error bars show the SEM. Statistical significance is determined by one-way ANOVA with Sidak correction for multiple comparisons. **d** Left: I−V plot from patch experiments on GIE from stage I to stage V. Right: Whole-cell conductance calculated at −100 mV on GIE from stage I to stage V with (orange) or without (blue) 100 μM NPPB. Circles indicate the number of independent experiments and error bars show the SEM. Statistical significance is determined by a Mann−Whitney test at each gametocyte stage and by ANOVA test for trend between stage 1 and stage 5, $p = 0.0013$, slope −134.7 ± 40.03 pS/stage transition. ****$p < 0.0001$, ***$p < 0.001$, **$p < 0.01$, *$p < 0.05$, ns: non-significant difference.

Fig. S5). First, we validated that Fluo-DHA was detectable in the gametocyte cytoplasm after 2 h incubation and exhibited anti-plasmodial activity against early gametocytes as assessed by a luciferase-based viability assay (Supplementary Fig. S5). Flow cytometry quantification showed that Fluo-DHA was taken up by stage II GIE, whereas uninfected erythrocytes were all negative (Fig. 4a, b and Supplementary Fig. S6). The pharmacological

specificity of this probe was validated by a competition assay with an excess of DHA (Fig. 4c). Importantly, preincubation with the NPP inhibitors NPPB and furosemide induced a significant decrease in the mean fluorescence intensity, indicating that Fluo-DHA uptake is partly mediated by NPPs in early GIE (Fig. 4d). About 50% of fluorescence remained upon NPPB treatment, suggesting that Fluo-DHA uptake may also partly occur by

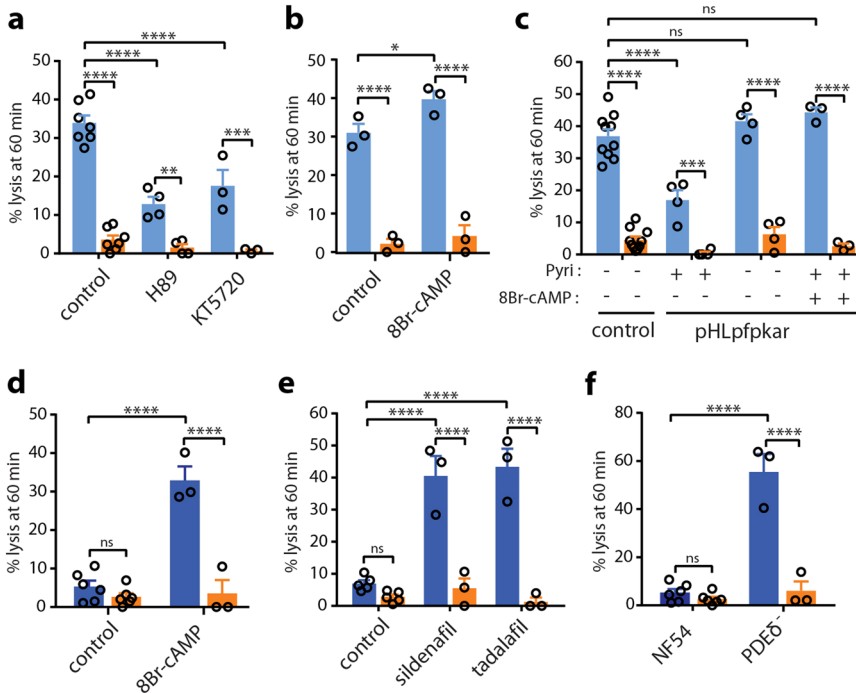

**Fig. 3 NPP activity is regulated by cAMP-signaling. a, b** Sorbitol-induced isosmotic lysis of stage II GIE with 100 μM H89 or 10 μM KT5720 (**a**), or with 100 μM 8Br-cAMP (**b**). **c** Sorbitol-induced isosmotic lysis of stage II GIE from the NF54 isolate (Control) and the transgenic pHL*pfpkar* line, cultivated with or without pyrimethamine (Pyri), or preincubated with 100 μM 8Br-cAMP. **d, e** Sorbitol-induced isosmotic lysis of stage V GIE with 100 μM 8Br-cAMP (**d**), sildenafil (**e**) or tadalafil (**e**). **f** Sorbitol-induced isosmotic lysis of stage V gametocytes from the NF54 isolate and the transgenic line PDEδ⁻. All experiments were performed in the presence (orange) or absence (blue) of 100 μM NPPB. Circles indicate the number of independent experiments and error bars show the SEM. Statistical significance is determined by one-way ANOVA with Sidak correction for multiple comparisons. ****$p < 0.0001$, ***$p < 0.001$, **$p < 0.01$, *$p < 0.05$, ns: non-significant difference.

diffusion through the erythrocyte membrane or by another route. Moreover, the viability at 48 h of early GIE following a 3-h pulse exposure to 150 nM Fluo-DHA or 700 nM artemisinin was slightly but significantly increased upon preincubation with NPPB, suggesting that NPP inhibition impacts the transport of these antimalarials into the infected erythrocytes (Fig. 4e). These results indicate that NPPs facilitate the uptake of artemisinin derivatives, and substantiate the idea that the slowdown of NPP activity in mature stages may account for their refractoriness to these drugs. To address this hypothesis, we evaluated Fluo-DHA uptake in synchronous cultures of stage II, III, IV, and V gametocytes. As observed for the decline in NPP activity, Fluo-DHA uptake also decreased during gametocyte maturation (Fig. 4f and Supplementary Fig. S6). The mean fluorescence intensity in GIE slowed down in stage III and reached the same level as that of NPPB-treated cells in stage V GIE. This profile is consistent with the increased IC₅₀ of Fluo-DHA in mature gametocytes (Supplementary Fig. S5) and correlates with the previously described chemosensitivity shift occurring at the transition from stage III to stage IV[28,32]. Therefore, the decline in NPP activity and in drug uptake along gametocytogenesis parallels the decrease in gametocytes sensitivity to several antimalarials[4,28]. Although we cannot rule out that the decrease in Fluo-DHA activity in mature stages reflects the slowdown of hemoglobin breakdown required for the activation of artemisinin[33], our results provide evidence that NPPs significantly contribute to the uptake of artemisinin derivatives in GIE.

**cAMP-mediated reactivation of NPPs increases uptake of artemisinin**. Next, we analyzed whether the uptake of artemisinin derivatives is also regulated by cAMP-signaling. In early GIE, we

observed that both pharmacological and genetic inhibition of PKA activity decreased the mean fluorescence intensity of Fluo-DHA in treated cells (Fig. 4g, h), suggesting that PKA activity facilitates Fluo-DHA uptake in immature GIE. Thus, we hypothesized that a reactivation of NPP activity in mature GIE may increase their susceptibility to these drugs. First, we addressed whether activating the cAMP pathway may enhance drug uptake by stage V GIE. Preincubation of mature GIE with the PDE inhibitors sildenafil and tadalafil strongly increased the mean fluorescence intensity of Fluo-DHA in treated cells, reaching the level observed in early GIE (Fig. 4i and Supplementary Fig. S7). The uptake of Fluo-DHA by mature GIE was significantly increased with 10 μM tadalafil, which approximately corresponds to tenfold the reported peak serum concentration reached in humans after 60 min following 20 mg oral dose ($C_{max}$ 378 ng/ml)[34]. As expected, preincubation of stage II GIE with tadalafil did not trigger any significant effect (Supplementary Fig. S7). To investigate the gametocytocidal effect of combining PDE inhibitors and artemisinin, we performed a gamete egress assay based on the ability of functionally viable mature gametocytes to undergo a temperature-dependent release of gametes from their host erythrocytes[35]. We observed that the incubation of mature GIE with a combination of 5 μM artemisinin and 30 μM tadalafil drastically reduced the proportion of egressed gametes compared to tadalafil or artemisinin alone, indicating that phosphodiesterase inhibitors potentiate the effect of artemisinin in mature gametocytes (Fig. 4j). Therefore, these results show that cAMP-mediated reactivation of NPPs in mature stages enhances the uptake of artemisinin, and importantly this mechanism can increase mature gametocyte sensitivity to this antimalarial.

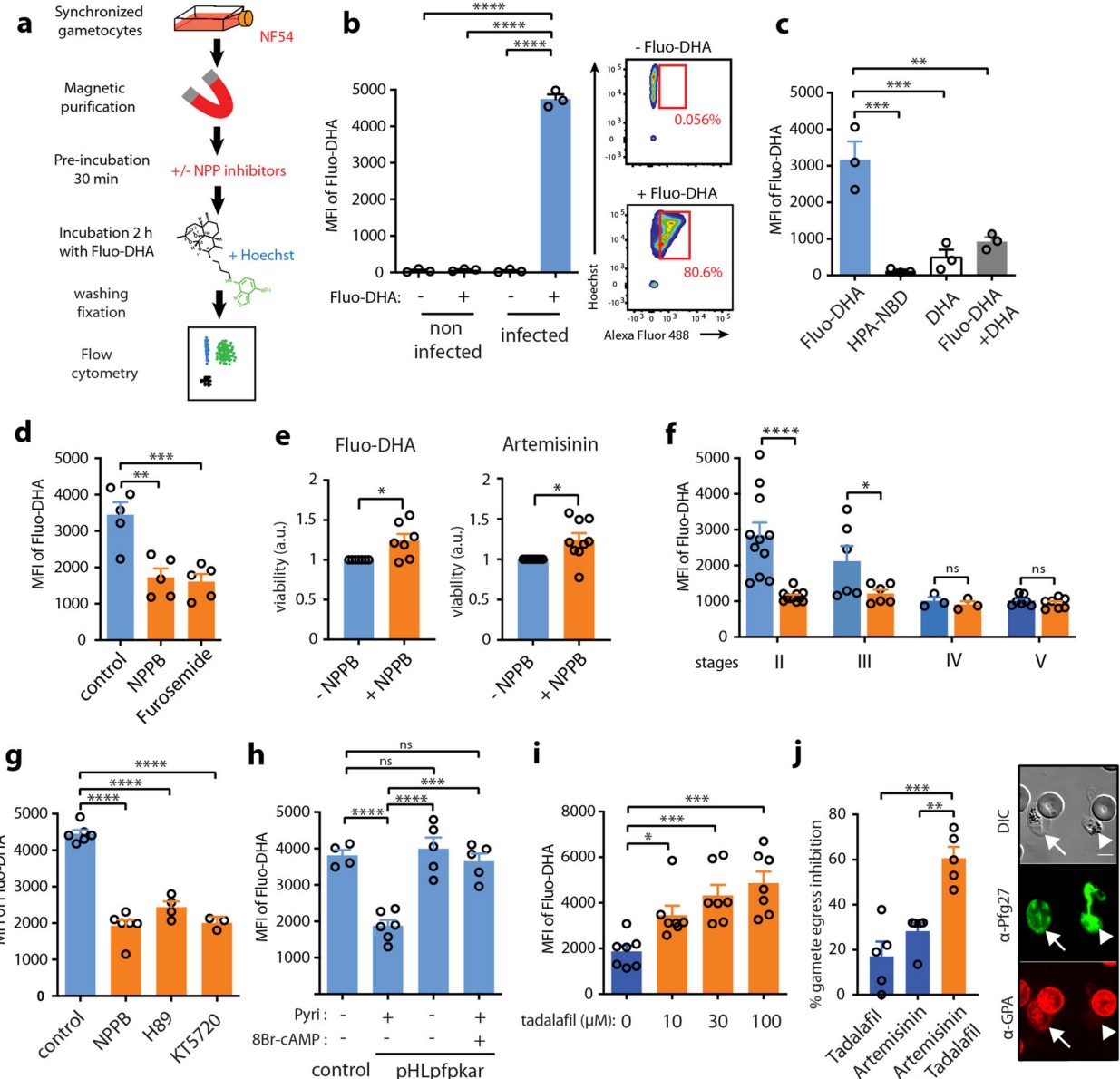

**Fig. 4 NPPs contribute to the uptake of artemisinin derivatives. a** Diagram illustrating the uptake assay. **b** Left: Quantification of Fluo-DHA uptake in uninfected erythrocytes and in early GIE by flow cytometry. Right: scatter plots showing the gating strategy for Fluo-DHA uptake. **c** Competition assay between Fluo-DHA and DHA, and control of HPA-NBD and DHA-induced fluorescence levels. **d** Inhibition of Fluo-DHA uptake in early GIE upon 100 μM NPPB or Furosemide incubation. **e** Viability (luciferase activity) of early gametocytes of the NF54-cg6-pfs16-CBG99 line 48 h after a 3-h incubation with 150 nM Fluo-DHA or 5 μM artemisinin, with or without 100 μM NPPB. The graph shows the ratio of luciferase activity (drug-treated/control) and is normalized to the condition without NPPB. a.u. arbitrary units. **f** Quantification of Fluo-DHA uptake during gametocytogenesis with (orange) or without (blue) 100 μM NPPB. **g** Fluo-DHA uptake in early GIE upon 100 μM H89 or 10 μM KT5720 incubation. **h** Fluo-DHA uptake in early GIE from NF54 (Control) and the transgenic pHL*pfpkar* line, cultivated with or without pyrimethamine (Pyri), or preincubated with 100 μM 8Br-cAMP. **i** Fluo-DHA uptake in stage V GIE upon 0, 10, 30 and 100 μM tadalafil incubation. **j** Left: % inhibition of gamete egress after a 24-h incubation with 5 μM artemisinin, with or without 30 μM tadalafil. Right: gamete egress observed by IFAs. Samples were co-stained with mouse anti-glycophorin A (GPA, red) and rabbit anti-Pfg27 (green) IgG. DIC differential interference contrast. Arrow: mature GIE with an intact erythrocyte membrane, arrowhead: egressed gamete. Scale bars: 5 μm. Circles indicate the number of independent experiments and error bars show the SEM. Statistical significance is determined by one-way ANOVA with Dunnet correction (**b−d**, **g**, **i**) or with Sidak correction (**f**, **h**, **j**) for multiple comparisons or by a Mann−Whitney test (**e**). ****$p < 0.0001$, ***$p < 0.001$, **$p < 0.01$, *$p < 0.05$, ns: non-significant difference.

## Discussion

It has been known for long that *P. falciparum* asexual stages modify their host erythrocyte to make them more permeable to supplementary nutrient uptake from the plasma and for removal of toxic waste by activating NPPs in the erythrocyte membrane. In this study we unravel how these NPPs are regulated during sexual parasite development and we propose that activating these pathways may facilitate artemisinin uptake by mature gametocytes. Our results raise a new model (Fig. 5) where NPP activity in immature GIE depends on the PKA-mediated phosphorylation of one or several proteins directly or indirectly involved in channel activity. According to this model, NPPs contribute to the uptake

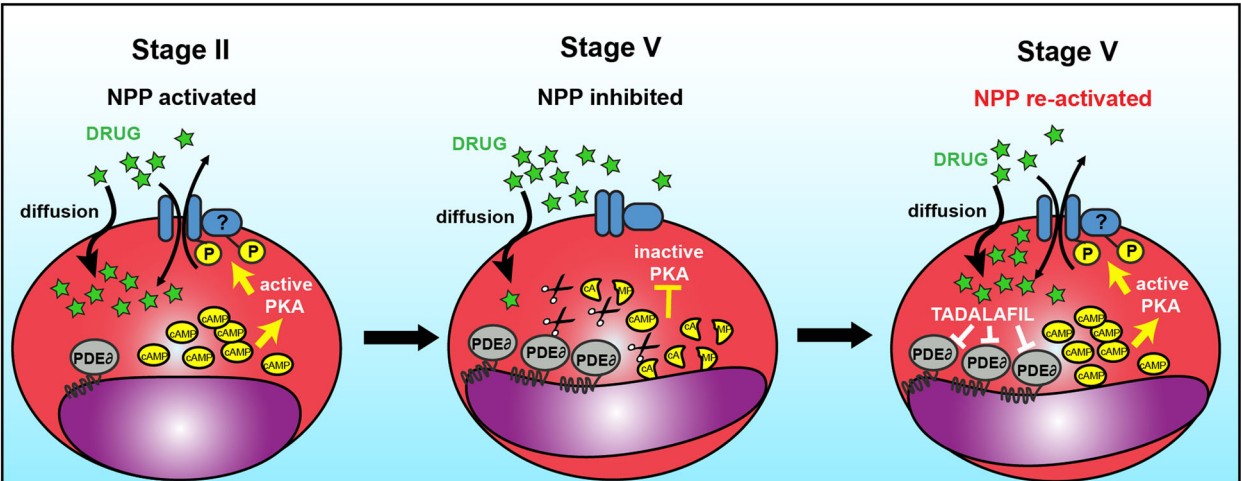

**Fig. 5 Model for cAMP-mediated regulation of NPP activity.** Left: In early GIE, PfPDEδ expression is low, resulting in high cAMP levels in the parasitophorous vacuole and in the host cell, thereby activating the human PKA. PKA phosphorylates one or several proteins directly or indirectly involved in NPP activity that contributes to the uptake of drugs. Middle: In mature GIE, PfPDEδ is highly expressed and degrades cAMP, leading to a decrease in PKA phosphorylation and NPP activity. Right: In mature GIE, inhibition of PfPDEδ by tadalafil results in increased levels of cAMP that reactivate PKA and NPPs, thereby restoring uptake of drugs.

of artemisinin and derivatives by immature GIE. Then in mature GIE, PfPDEδ, whose catalytic domain is predicted to be exposed outwards of the parasite[36], is highly expressed and degrades cAMP[26]. The resulting drop of cAMP level in the parasitophorous vacuole and in the host cell may reduce the phosphorylation of these proteins and consequently decrease NPP activity and drug uptake. Importantly, pharmacological inhibition of PfPDEδ can reactivate NPP activity and restore uptake of artemisinin derivatives by mature gametocytes, leading to an increased susceptibility to these drugs. Altogether these results show that *P. falciparum* gametocytes regulate infected erythrocyte permeability and that this permeability may facilitate uptake of some antimalarials.

Our results showing that NPPs are still active in immature GIE contradict the paradigm that NPP activity is abolished in sexual stages. This dogma was based on a single study showing refractoriness of GIE to lysis upon short exposition to isosmotic sorbitol solution[17]. In contrast, our data report that GIE were lysed after longer exposition to sorbitol, indicating the presence, although slower than in trophozoites, of NPP activity in immature GIE. In support of the existence of NPPs in sexual stages, recent RNA sequencing analyzes highlighted that sexually committed parasites show higher transcription of *RhopH* and *clag* genes[37], whose products are involved in NPP activity[12–14]. These observations suggest that members of the RhopH complex synthesized in sexually committed parasites may be discharged upon egress onto the membrane of the erythrocyte targeted for invasion, within which a gametocyte will develop. Accordingly, our data show the presence of at least one member of the RhopH complex in GIE. This hypothesis would be consistent with the fact that gametocytes should absorb nutrients from the extracellular medium, such as panthotenate or isoleucine, for which NPPs is the major route[21]. In addition, NPPs have been proposed to play a role in exporting amino acids liberated by the digestion of hemoglobin from the infected erythrocyte, thereby protecting the cell against the osmotic challenge posed by elevated intracellular amino acid levels[20]. Thus, the slowdown of NPP activity in mature GIE may result from the completion of hemoglobin digestion in these stages[38]. Absence of channel activity in mature stages may allow the infected erythrocyte to decrease its osmotic fragility and thus to persist longer in the blood circulation. This process appears to be tightly regulated by cAMP-signaling pathway, as previously observed for asexual stages[16]. These results are in accordance with the detection of CLAG3/RhopH1, one of the major component of NPPs, in a global phosphoproteomic analysis of *P. falciparum* parasites[39]. As part of the cAMP-signaling cascade, the PfPDEδ enzyme, whose expression increases in mature gametocytes[26], plays an instrumental role in the NPP decline. Interestingly, this enzyme has also been reported to govern the switch in GIE deformability that enables mature gametocytes to circulate several days in the bloodstream and avoid clearance by the spleen[26,40,41]. Therefore, by decreasing GIE osmotic fragility and increasing GIE deformability, PfPDEδ appears to be the key regulator of mature gametocytes persistence in the blood circulation. Importantly, our present work suggests that targeting this enzyme with the FDA-approved drug tadalafil may enhance artemisinin uptake by mature gametocytes, thereby decreasing their refractoriness to this antimalarial.

As PDE inhibitors also render mature GIE rigid and hence may promote their clearance by the spleen[26], they represent novel drug leads potentially capable of blocking malaria transmission by impacting on both gametocytes circulation and susceptibility to artemisinin derivatives.

## Methods

**Parasite culturing and gametocyte production.** The *P. falciparum* NF54 strain, the B10 clone and the transgenic lines pHL*pfpkar*, *PfPDEδ*−, NF54-cg6-pfs16-CBG99 and NF54-pfs47-pfs16-GFP have been described elsewhere[16,23,27,42]. Parasites were cultivated in vitro under standard conditions using RPMI 1640 medium supplemented with 10% heat-inactivated human serum and human erythrocytes at a 5% hematocrit. To obtain synchronous asexual stages, parasites were synchronized by the isolation of schizonts by magnetic isolation using a MACS depletion column (Miltenyi Biotec) in conjunction with a magnetic separator, and placed back into culture. After invasion of merozoites, a second magnetic isolation was used for the selection of ring-stage parasites to obtain a tighter window of synchronization. Synchronous production of specific gametocytes stages was achieved by treating synchronized cultures at the ring stage (10–15% parasitemia, day 0) with 50 mM *N*-acetylglucosamine (NAG) for 5 days to eliminate asexual parasites. Gametocyte preparations were enriched in different experiments by magnetic isolation. Stage I GIE were collected at day 1 after initiating NAG treatment, stage II GIE were collected at days 2 and 3, stage III GIE were collected

at days 4 and 5, stage IV GIE were collected at days 6 and 7, and stage V GIE were collected at day 8 onwards.

**Isosmotic lysis.** 500 µl of *P. falciparum* cultures (containing gametocytes or asexual stages) with a parasitemia >0.5% were washed once with RPMI and incubated in 1.5 ml tubes for 60 min at 37 °C in 500 µl isosmotic solution containing either 300 mM sorbitol, 300 mM Alanine or 150 mM PhTMA$^+$ supplemented with 10 mM Hepes, 5 mM glucose and with a pH adjusted to 7.4. For each experiment, five tubes were prepared including one tube containing 100 µM NPPB, Furosemide, Ro5-4864, or PK11195 (all purchased from Sigma-Aldrich). At each sampling time, one tube was centrifuged and smears were prepared and stained with Giemsa R solution (RAL diagnostics). Parasitemia was estimated for each point by counting infected cells out of at least 4000 erythrocytes, and lysis percentage was calculated using the formula: % lysis $(t) = [1 - (\text{parasitemia } (t)/ \text{parasitemia } (t0))] \times 100$.

Microscopy examination of Giemsa-stained smears allowed to morphologically confirm that only gametocytes, but not contaminating asexual parasites, were counted. The percentage of lysis of stage I GIE was determined using the NF54-pfs47-pfs16-GFP line[24] by fluorescence microscopy at ×40 magnification on a Leica DMi8 microscope. Addressing the role of cAMP-signaling in isosmotic lysis, stage II GIE (days 2 and 3 post NAG treatment) were preincubated 30 min with 100 µM H89, 10 µM KT5720 or 100 µM 8Br-cAMP, and mature GIE (days 8–11 post NAG treatment) were preincubated 30 min with tadalafil or sildenafil at different concentrations (from 10 to 100 µM). All inhibitors were purchased from Sigma-Aldrich or Euromedex.

For standard semi-quantitative isosmotic lysis assays on uninfected erythrocytes, hemoglobin release was used to estimate lysis. Erythrocytes were washed three times in culture medium without serum and resuspended at 50% hematocrit. Time courses started with the addition of a 10 µl packed cells suspension to 1 ml of the sorbitol isosmotic solution. Experiments were performed in triplicate. At predetermined intervals (0, 15, 30, 60 min), microcentrifuge tubes were centrifuged for 30 s and 200 µl of the supernatant solution was transferred into 96-well plates for spectrophotometric estimation of hemoglobin concentration by absorption at a wavelength of 540 nm (A540). In all experiments, the A540 value corresponding to full hemolysis of erythrocytes was estimated from the final A540 value achieved in the supernatant solution from a suspension where 5 µl of Triton X-100 is added. Data analyses were carried out as previously described.

**Patch-clamp.** Patch-clamp experiments were performed at room temperature using the whole-cell configuration. Pipettes were pulled using a DMZ Universal Puller (Zeitz Instruments, Germany) from borosilicate glass capillaries (GC150F-10, Harvard apparatus, UK) to obtain a tip resistance of 10–15 MΩ. Pipette solution was: 145 mM KCl, 1.85 mM CaCl$_2$, 1.2 mM MgCl$_2$, 5 mM ethylene glycol-bis(β-aminoethyl ether)-N,N,N′,N′-tetraacetic acid (EGTA), 10 mM Hepes, 10 mM, pH 7.2. Bath solution was: 145 mM NaCl, 5 mM KCl, 1 mM CaCl$_2$, 1 mM MgCl$_2$, 10 mM Hepes, 10 mM glucose, pH 7.4. Seal resistance were 4–20 GΩ, and whole-cell configuration was obtained by a brief electrical pulse (zap) and was assessed by the development of small capacitance transient currents and reduction of access resistance. Whole-cell current were recorded using either an Axopatch 200B (Molecular Devices, USA) or a RK400 (Biologic, France) amplifier, with voltage command protocols and current analysis done with pclamp10 suite software (Molecular Devices, USA) or WinWCP4.7 software (J. Dempster, Strathclyde University, UK), respectively. Currents were elicited by generating a series of membrane potentials from +100 to −100 mV in −10-mV steps for 500 ms, from a holding potential of 0 mV. For stage I GIE, the gametocyte fluorescent NF54-pfs47-pfs16-GFP line[24] was used to easily distinguish by microscopy between asexual and early sexual stages.

**Fluorescence microscopy.** To visualize the uptake of Fluo-DHA, GIE were incubated 2 h with Fluo-DHA or dimethyl sulfoxide (DMSO) 0.1% at 37 °C. Cells were stained with Hoechst 33342 (1/20,000, Thermo Fisher Scientific) for 10 min at 37 °C. After one wash with phosphate-buffered-saline (PBS) 1×, cells were fixed for 10 min at room temperature with 1% paraformaldehyde (PFA) (Electron Microscopy Science) and 0.25% glutaraldehyde (Sigma-Aldrich) in PBS 1×. After three washes with PBS 1×, infected cells were observed on a glass slide at ×100 magnification using a Leica DMi8 microscope.

For immunofluorescence analysis, smears were prepared with NF54 culture containing immature GIE and asexual stages. Slides were air-dried, fixed for 5 min in 75% acetone/25% methanol at −20 °C, washed three times in PBS, preincubated for 2 h in 1× PBS/2% bovine serum albumin, incubated overnight with a mouse antibody directed against RhopH2 (1/50)[43] and then 1 h with a rabbit antibody raised again the Pf11-1 protein (1/5,000)[44]. Then, slides were incubated with Alexa Fluor 488-conjugated goat anti-rabbit (1/2,000), Alexa Fluor 594-conjugated goat anti-mouse antibody (1/2,000) and Hoechst 33342 (1/20,000) (Thermo Fisher) for 1 h at room temperature. All samples were observed at ×100 magnification using a Leica DMi8 microscope. The RhopH2-specific mouse serum was obtained from C. Braun-Breton and the Pf11-1-specific rabbit serum was obtained from O. Mercereau-Puijalon.

**Quantification of Fluo-DHA uptake assay.** The synthesis of the Fluo-DHA probe was described in Sissoko et al.[31]. Quantification of Fluo-DHA uptake was performed using flow cytometry. GIE were preincubated or not with 100 µM NPPB or with 100 µM Furosemide for 30 min and then incubated with 1 µM of Fluo-DHA for 2 h at 37 °C, 5% CO$_2$ and 5% O$_2$. To address the role of cAMP-signaling in Fluo-DHA uptake, early GIE (day 2 post NAG treatment) were preincubated 30 min with 100 µM H89, 10 µM KT5720 or 100 µM 8Br-cAMP, and mature GIE (days 8–11 post NAG treatment) were preincubated 30 min with tadalafil or sildenafil at different concentrations (from 10 to 100 µM). To assess the specificity of Fluo-DHA uptake, early GIE were preincubated or not with 20 µM DHA for 2 h and then incubated with Fluo-DHA (1 µM) for 2 h. DHA alone (20 µM) and HPA-NBD (1 µM) were used as control. Twenty minutes before the end of incubation, GIE were stained with Hoechst 33342 (1/10,000). Cells were then washed with PBS 1× and fixed for 10 min at room temperature with PBS, 1% PFA and 0.25% glutaraldehyde. The percentage of Fluo-DHA-positive cells was quantified using Fortessa (BD Biosciences) cytometer.

**Gametocyte survival assay.** To calculate the IC$_{50}$ for Fluo-DHA on early or mature gametocytes, $2 \times 10^5$ MACS-purified early GIE (day 2 post NAG treatment) or mature GIE (days 8–11 post NAG treatment) from the NF54-cg6-Pfs16-CBG99 line were incubated with serial dilutions of Fluo-DHA for 3 h, then GIE were washed and incubated with complete medium without drugs for 72 h. To perform viability assays with drugs and inhibitors, early GIE were preincubated at 37 °C for 30 min with 100 µM NPPB or 0.1% DMSO in complete medium and then incubated with the same inhibitors supplemented with 700 nM or 5 µM artemisinin (Sigma-Aldrich), 150 nM Fluo-DHA or 0.1% DMSO for 3 h. GIE were then washed and incubated with complete medium without inhibitors and drugs for 48 h. Cell viability was evaluated by adding a non-lysing formulation of 0.5 mM D-Luciferin substrate[23] (Sigma-Aldrich) and measuring luciferase activity for 1 s on a plate Reader Infinite 200 PRO (Tecan®). All experiments were performed in triplicate on 96-well plates.

**Gamete egress assay.** NF54 cultures containing 2−5% stage V gametocytes were incubated for 24 h at 37 °C with 5 µM artemisinin supplemented or not with 30 µM tadalafil. Controls comprising gametocytes exposed to the same final concentration of DMSO (0.1%) were processed in parallel. After the incubation, 100 µl samples from each condition were pelleted at $1000 \times g$ for 1 min and rapidly resuspended in 50 µl human serum at room temperature for 30 min during which gametogenesis took place. Samples were then pelleted at $1000 \times g$ for 1 min, smeared on glass slides, methanol-fixed and co-stained with mouse anti-glycophorin A (GPA, 1/1000, Santa Cruz Biotechnology) and rabbit anti-Pfg27[45] (1/2000) followed by anti-rabbit Alexa 488- and anti-mouse Alexa 594-conjugated IgG (1/2000, Life Technologies). Samples were observed at ×100 magnification using a Leica DMi8. At least 100 GIE were analyzed for each sample. Percent gamete egress was determined by calculating the % of GPA-negative round gametes in the total GIE population detected by Pfg27 staining.

**Statistics and reproducibility.** Group data are presented as mean ± s.e.m. Statistics for all datasets were performed using GraphPad Prism. Statistical significance is determined by one-way ANOVA with Dunnet correction or with Sidak correction for multiple comparisons or by a Mann−Whitney test or paired $t$ test. Sample sizes and replicate details are described in the relevant figure descriptions.

## Data availability

All experimental data generated or analyzed during this study are included in this published article (and its Supplementary Information files: Supplementary Information and Source Data).

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

## Acknowledgements

The authors thank D. Baker (LSHTM) for providing the *PfPDEδ−* line, C. Braun-Breton for providing the anti-RhopH2 antibodies and O. Mercereau-Puijalon for providing the anti-Pf11-1 antibodies. The authors acknowledge T. Guilbert at the imaging core facility Imag'IC and the Flow Cytometry core facility CYBIO of the Institut Cochin for technical help. This study was supported by grants from Laboratory of Excellence GR-Ex, reference ANR-11-LABX-0051. The labex GR-Ex is funded by the IdEx program "Investissements d'avenir" of the French National Research Agency, reference ANR-18-IDEX-0001. C.L., D.B., F.D., A.M., M.-E.N., G.N., and L.B. acknowledge the financial support from the Cnrs, Inserm and the Fondation pour la Recherche Médicale ("Equipe FRM" grant EQ20170336722).

## Author contributions

G.B., S.E. and C.L. conceived the project. G.B., D.B., J.C., R.D., S.E. and C.L. designed and interpreted the experiments. G.B., D.B., F.D., A.M., A.S., M.-E.N., G.N., L.B., N.K., D.R. and C.L. performed the experiments. S.H., G.S., P.A., R.M.M., J.-J.L.-R. and R.D. contributed resources or data. C.L. wrote the article, with major input from G.B., J.C., R.D. and S.E.

## Competing interests

The authors declare no competing interests.
