## [Peer Review File · Communications Biology]

Reviewers' Comments:

Reviewer #1:

Remarks to the Author:

This is a well-performed study that thoroughly evaluates important questions: the extent to which gametocytes modify the permeability of their host erythrocyte, the mechanism by which they do so, and how this changes over the course of gametocyte development. The authors evaluate these effects by two complementary approaches and empirically show that their work has relevance in potentially allowing the enhancement of activity of antimalarial drugs against parasite transmissive stages. This work will without question influence the field both in terms of basic gametocyte biology and drug development.

I have some overarching questions, though I do not expect the answers to them to challenge the overall messages of this work:

These assays seem to expose cells to sorbitol (/other osmotic agents) for substantially longer than in previous work. It would be helpful if the authors could comment on whether any significant lysis of uninfected erythrocytes occurs in this time. If this does occur it would not discredit the effects the authors have observed but would change the interpretation of the absolute values measured for lysis.

In the DHA experiments it is unclear to me if "percentage of DHA positive cells" is the best metric. It seems to be creating a binary classification where one is not really expected. Average fluorescence seems like a better metric to me. If the authors considered this and dismissed it, it would be interesting to hear their reasoning. It is unclear if the figures 70.8 and 31.9 in Fig S5 are calculated excluding the uninfected cells in the lower left quadrant – it seems to me they should be? In general, especially if the authors proceed with their current analysis method, it would be helpful if flow cytometry plots for all flow-cytometry data presented were presented in Supplemental.

When the authors write 'Stage I GIE were collected at day 1 post NAG treatment'(, etc.) do they mean after the start or end of the treatment? If this were the start it would be useful to know whether there was any remaining asexual contamination in for instance the stage II GIE culture, and whether this was excluded, since a decline in contaminating asexual parasitaemia could drive some of the effects seen (I accept the authors have tried to control for this at Stage I by staining).

While the raw data is generally visible in the figures, as indicated in the data availability statement, it is not for Fig 4d, or for some of the supplemental figures. It would be good to include this data, ideally with individual points on the barchart as in the other figures, but if not, as supplemental tables.

Throughout: There is some confusion to the reader about whether the term NPP is used as a singular or a plural item. My suggestion to maximise readability would be to use 'NPPs' throughout (e.g. line 55 changes from 'full identity of the NPP has never' to 'full identity of the NPPs have never'.) This form appears to have been used in at least some past work.

Small things:

Line 58: change to 'the level of NPP activity'

Line 59: 'Refractoriness of GIE to lysis upon short exposition to isosmotic sorbitol solution has led to the dogma that NPP are totally absent in gametocyte stages (17)' – this feels unfair to the cited paper which concludes with 'More sophisticated experiments will be required to see if gametocytes are totally lacking this pathway or whether it exists but at an activity too low to induce lysis'. It is possible that this has indeed become dogma, but if so a further review reference would be helpful

(also in discussion).

Line 60: 'exposition' to exposure

Line 70: 'the/a cyclic AMP signalling cascade'

Line 107: 'leading to currents in mature GIE corresponding to levels usually recorded in uninfected erythrocytes'.

Line 149: Change 'embedded' to 'taken up' or similar.

Line 167: to "gametocytes' sensitivity" or to "gametocyte sensitivity"

Line 175: change to 'activating the cAMP pathway '

Line 184: Fig ref should be to 4j. The authors could consider modelling whether this effect is statistically significantly greater than an additive effect (eyeballing it this seems likely), though these methods can be contentious.

Line 192: 'interfering' can be read as downregulating NPPs, when the authors mean the opposite to facilitate drug uptake

Line 207: change to 'In support of the existence of NPPs'

Line 211: Do the authors mean invasion not egress?

Line 226: delete 'the' in 'clearance from the spleen'.

Line 229: It would be helpful if the authors could comment on whether the concentrations of tadalafil they have identified as relevant are likely to be achievable in patient serum. My back-of-the-envelope concentrations suggest they may not (but may well be wrong). Even if they are not this point can be raised, but the caveat should be noted.

Line 257: spinned - > centrifuged

Line 301: do the authors mean Fluo-DHA positive not GFP-positive?

Line 303. 2x10 not 2.10

Reviewer #2:

Remarks to the Author:

The manuscript by Bouyer et al analyses the ability of early and late sexual stage gametocytes to be permeable to isotonic solutes and Fluo-DHA when NPP blockers or drugs that regulate cAMP and hence activity of PKA are added to the culture medium. The authors propose that early stage gametocytes have functional new permeation pathways because of low PDE expression but that the nutrient channel is inhibited in mature gametocytes when PDE levels are high as levels of PKA phosphorylation, required for NPP activation, is low.

There are several concerns I have with the manuscript, mostly centred around the use of controls and the high concentrations of drugs that are used throughout. Consequently, it is difficult to assess whether the results they are seeing are truly meaningful and one can't be sure if the effects are a direct or indirect effect on NPP function. Whilst the RhopH products are transcribed in committed schizonts, it is important to demonstrate whether the RHopH proteins are actually expressed on early gametocytes. Are the authors proposing the NPPs differ between asexual stage and early sexual stages because the difference in sorbitol sensitivity is very large and moreover, artemisinin is not taken up by the NPP in the asexual stages.

More specific comments are below:

Figure 1. Unlike trophozoite-stage parasites which rapidly lyse after 15 mins incubation in sorbitol, gametocyte sorbitol lysis was only double that of late rings. For this assay, how can the authors be confident that the lysis they are observing is not due to a small number of contaminating asexual stages?

Why was the NPP assay assessed via Giemsa staining of parasites, rather than measuring RBC lysis in a spectrometer? I was surprised the authors didn't use the GFP line for these experiments to determine what the percentage of GIE were in each sample.

In part (a), where n=3, is this technical or biological repeats?

Are the small changes in membrane conductance at -100mM in Fig (d) meaningful. How does this

compare to ring-stage parasites for example, which don't have NPP? A comparison to ring stages would have been a very useful control.

For Fig 1f: It was not clear how viability was measured via luciferase (?measure release of luciferase into the culture medium). This may also be caused by parasites being more fragile. If viability was affected, washout experiments would show fewer transition to later stage gametocytes, was this observed?

Figure 2: Again, rings and trophs would have served as very useful controls on the graphs. None of the figures show uninfected erythrocytes (see line 108) but rings serve as better negative control.

Figure 3- the amount of H89 and 8-Br-cAMP used in this experiment seems very high at 100 microM. What concentrations have these PKA inhibitors have been used in other studies? It would have been good to do some titrations of these compounds and also to use asexuals as controls in these experiments.

Figure 4. The text on Ln 123-4 mentions revertant parasite line that has shed the overexpressing episome - was this shown as removing the drug doesn't necessarily remove episomes which can be very stable in parasites?

What happens when the PDE inhibitors sildenafil or tadalafil is used on Stage 1 and II gams or an asexual stage parasites? If cAMP levels are high in the early gams then then the PDE inhibitors would not further increase isosmotic lysis. It would have been good to see this as a control. DHA does not get taken up by the NPP in troph stages, so how does one envisage that the NPP may be used in early gametocytes but not in the asexual stages? It would have been good if trophs were used in this experiment as a negative control for these experiments. The amount of artemisinin used in the assay is very high, did you try any titration experiments? Likewise, there is no effect with Tadalafil at the 10 micromolar concentration.

The cells incubated with Fluo-DHA were only washed once prior to fixation so this may explain why the %Fluor-DHA positivity is so high. Again, how does a ring or trophozoite-stage parasite compare?

Supp Fig 4 – How does the IC50 of Fluo-DHA change going from early gametocytes to late gametocytes?

Reviewer #3:

Remarks to the Author:

In their manuscript "Plasmodium falciparum sexual parasites regulate infected erythrocyte permeability" Bouyer et al. investigate the presence of NPPs in Plasmodium falciparum gametocytes. They use a variety of techniques, including patch clamping, imaging and survival assays to determine if the NNPs are active in the sexual stage of the parasite. Furthermore, they investigate whether artemisinin requires the NPP to enter the infected erythrocyte. In their final experiments, the authors use clever genetic experiments coupled with the use of inhibitors and activators to show that the activity of the NNP requires the activity of the parasite kinase PKA. The authors conclude with a model about the regulation of the NPP by PKA. The authors represent the data clearly (the color coding throughout makes the figures easy to follow) and provide good controls throughout. There are several novel aspects to these findings, but not all of the experiments are completely convincing. There are several additional experiments that the authors might consider to strengthen their findings.

-The authors describe how the gametocytes culture is established and purified, but do not show any images of the resulting gametocyte cultures. From the materials and methods section, it appears that the authors do not induce gametocytogenesis through metabolic depletion with spent medium or lysophosphatidylcholine depletion, but instead solely rely on the removal of asexual

parasites using extended NAG treatment. Hence, the starting parasitemia of gametocytes is likely to be very low. As NAG treatment does not remove asexual stage parasites within one cycle (Miao et al. <http://dx.doi.org/10.1016/j.exppara.2013.09.010>), there are likely to be asexual stage parasites in the gametocyte culture; with a low parasitemia of gametocytes, even if a low number of asexual parasites survive, these will likely still form a relatively large fraction of the population. Without knowing how pure the gametocyte cultures are, the lysis of parasites shown in Figure 1 and 2 could be explained by the presence of trophozoites or schizonts in the culture. As the level of asexual parasites will drop over time in the presence of NAG, this could explain the drop in the lysis in the culture over time. The authors should give an indication of the percentage of gametocytes in their culture that is achieved at each stage with their method, either by flow cytometry using the strain they described in which GFP is controlled by the pfs16 promoter or several large fields of purified parasites stained either with Giemsa or anti-Pfs16 (or similar gametocyte-specific antibody).

-Throughout, the authors rely on a relatively indirect measure (lysis of parasites) to investigate the presence of the NPP. In the introduction, they mention the role of the high molecular weight rhoptry complex in the induction of the NPP. A much more direct way to look at the presence of the NPP would be immunofluorescence microscopy using antibodies against members of this complex. The authors reference two papers that show that the high molecular weight rhoptry complex is present in schizonts in the introduction, but both papers describe asexual parasites; there is (to my knowledge) no information about the presence of this complex in schizonts that are committed to form gametocytes.

-The authors could also use PKG inhibitors to remove asexual parasites, which may act faster than NAG (Portugaliza HP et al. Reporter lines based on the gexp02 promoter enable early quantification of sexual conversion rates in the malaria parasite *Plasmodium falciparum*. doi: 10.1038/s41598-019-50768-y).

-Related to this, the authors use fluorescent derivatives of artemisinin to show uptake of compounds, but this is a rather non-standard assay and, as the authors acknowledge, if artemisinin uses the NPP, it is not the only pathway for the drug to enter the cells. A more common way to show uptake by the NPP is through the use of PPIX (Sigala PA, Crowley JR, Henderson JP, Goldberg DE. 2015. Deconvoluting heme biosynthesis to target blood-stage malaria parasites. *eLife* 4:e09143. doi: 10.7554/eLife.09143). This would allow for costaining with gametocyte markers or the visualization of GFP produced using a gametocyte-specific promoter. In conjunction with the NPP inhibitors that the authors use, the use of PPIX would show definitively that the NPP is active in gametocytes.

-The results in Figure 1D are hard to interpret without data on uninfected cells. What is the level of conductance in uninfected cells?

-The authors should mention that asexual ring-stage parasites, in which the NPP is not yet active, are sensitive to artemisinin. Hence, artemisinin can penetrate the erythrocyte without NPP and kill parasites.

-The PfpDE δ line that the authors use is a 3D7 derivative and hence may convert to gametocytes at a vastly different rate than the NF54 strain used by the authors. This caveat should be mentioned. With the advent of genetic manipulation of malaria parasites, a complemented PfpDE δ line could be made that would provide a better comparison (admittedly still a lot of work).

-The model that the authors propose in Figure 5 would benefit from additional details. The white arrow before 'active PKA' actually passes two membranes. As neither PKA nor cAMP can pass membranes and PKA is not thought to be exported, it is not clear how this step would occur. Other than the export of proteins and the formation of Maurer's clefts, no pathway for modification or signaling to the host cell has been described. The authors should provide an explanation how they

think this step takes place.

Reviewers' comments:

We thank the reviewers for their valuable suggestions and comments. In response to these we have carried out several new experiments and believe the manuscript is much improved as a result. We hope that the revised version will be suitable for publication following further review.

Reviewer #1 (Remarks to the Author):

This is a well-performed study that thoroughly evaluates important questions: the extent to which gametocytes modify the permeability of their host erythrocyte, the mechanism by which they do so, and how this changes over the course of gametocyte development. The authors evaluate these effects by two complementary approaches and empirically show that their work has relevance in potentially allowing the enhancement of activity of antimalarial drugs against parasite transmissive stages. This work will without question influence the field both in terms of basic gametocyte biology and drug development.

I have some overarching questions, though I do not expect the answers to them to challenge the overall messages of this work:

These assays seem to expose cells to sorbitol (/other osmotic agents) for substantially longer than in previous work. It would be helpful if the authors could comment on whether any significant lysis of uninfected erythrocytes occurs in this time. If this does occur it would not discredit the effects the authors have observed but would change the interpretation of the absolute values measured for lysis.

In the revised Figure 1, we included an experiment where we tested the hemolysis of uninfected erythrocytes in isosmotic sorbitol for 60 minutes. The hemolysis remains basal all long experience, with 0.45 +/- 0.23 % erythrocytes lysis at 15 min and 0.4 +/- 0.07 % at 60 min (n=3). With 100 μM NPPB, lysis at 60 min is 0.3 +/- 0.09 %, not statistically different from without NPPB. Therefore, these conditions have no or very little effects on hemolysis of uninfected erythrocytes. These results are consistent with the patch-clamp data on uninfected erythrocytes that we also included in the revised manuscript (revised Figure 1d).

Revised Figure 1

In the DHA experiments it is unclear to me if “percentage of DHA positive cells” is the best metric. It seems to be creating a binary classification where one is not really expected. Average fluorescence seems like a better metric to me. If the authors considered this and dismissed it, it would be interesting to hear their reasoning. It is unclear if the figures 70.8 and 31.9 in Fig S5 are calculated excluding the uninfected cells in the lower left quadrant – it seems to me they should be? In general, especially if the authors proceed with their current analysis method, it would be helpful if flow cytometry plots for all flow-cytometry data presented were presented in Supplemental.

We agree with the reviewer and we re-analyzed all the uptake experiments using Flow Jo software to calculate the mean fluorescence intensity (MFI) values. These analyses are now shown in the revised manuscript (Revised Figure 4 and Supplemental Figures S6 and S7).

Revised Figure 4

In Fig S5 (now revised Figure S6), uninfected cells were excluded from the analysis since gametocyte preparations were enriched by magnetic isolation before treatment with Fluo-DHA and then remaining uninfected cells were excluded using Hoechst staining (uninfected RBCs are Hoechst negative).

It would be very tedious to present all flow-cytometry plots, since for several conditions, experiments were performed up to 11 times. Statistical analysis are strong, so we do not think that showing all flow-cytometry data would be of interest for the scientific community. However, in the revised manuscript we now present the most representative flow cytometry plots for each gametocyte stage (from stage II to V) in supplemental data (revised Figure S6).

Revised Supplemental Figure S6

When the authors write ‘Stage I GIE were collected at day 1 post NAG treatment’(, etc.) do they mean after the start or end of the treatment? If this were the start it would be useful to know whether there was any remaining asexual contamination in for instance the stage II GIE culture, and whether this was excluded, since a decline in contaminating asexual parasitaemia could drive some of the effects seen (I accept the authors have tried to control for this at Stage I by staining).

Stage I GIE were collected at day 1 after the start of the NAG treatment. To exclude asexual contamination, we used the NF54-pfs47-pfs16-GFP reporter gene line, which expresses GFP selectively in gametocytes. Since only fluorescent parasites were analysed in isosmotic lysis and patch-clamp experiments, our results could not result from any contaminating asexual parasitemia.

While the raw data is generally visible in the figures, as indicated in the data availability statement, it is not for Fig 4d, or for some of the supplemental figures. It would be good to include this data, ideally with individual points on the barchart as in the other figures, but if not, as supplemental tables.

In our manuscript individual points were shown on the bar charts of all figures excepted for Figure 1d, 1e and 2d. In the revised manuscript we included individual points on the bar charts in revised Figure 2d. In Figure 1d and 1e, mean +/- sem was preferred for sake of clarity due to dispersion of several points. However figures including all data points are now available in revised Supplemental Figure S1.

Throughout: There is some confusion to the reader about whether the term NPP is used as a singular or a plural item. My suggestion to maximise readability would be to use 'NPPs' throughout (e.g. line 55 changes from 'full identity of the NPP has never' to 'full identity of the NPPs have never'.) This form appears to have been used in at least some past work.

Thanks for the suggestion, this has been modified in the revised manuscript.

Small things:

Thanks, these have been modified in the revised manuscript. We answer to specific questions below:

.Line 58: change to 'the level of NPP activity'

Line 59: 'Refractoriness of GIE to lysis upon short exposition to isosmotic sorbitol solution has led to the dogma that NPP are totally absent in gametocyte stages (17)' – this feels unfair to the cited paper which concludes with 'More sophisticated experiments will be required to see if gametocytes are totally lacking this pathway or whether it exists but at an activity too low to induce lysis'. It is possible that this has indeed become dogma, but if so a further review reference would be helpful (also in discussion).

We fully agree with the reviewer's comment with respect to the concluding sentence of the article by Saul et al. ..."*More sophisticated experiments will be required to see if gametocytes are totally lacking this pathway or whether it exists but at an activity too low to induce lysis*" and by the way, that's kind of the point of this article. Nevertheless this sentence is preceded by the sentence "*Secondly, it suggests that there is a marked difference in the metabolic requirements of maturing asexual stage parasites and gametocytes as the inability of sorbitol to lyse erythrocytes infected with the latter suggests that the high levels of the new anion permeation pathway present in membranes of erythrocytes infected with maturing trophozoites and schizonts are not present in those infected with gametocytes.*", which tends to imply that the metabolic needs of gametocytes are far below those of asexuals and thus do not required NPPs activity. Since then no study has revisited this assertion, which is perhaps not a dogma but in any case a fact accepted by the majority of parasitologists working on *Plasmodium falciparum*.

Line 60: 'exposition' to exposure

Line 70: 'the/a cyclic AMP signalling cascade'

Line 107: 'leading to currents in mature GIE corresponding to levels usually recorded in uninfected erythrocytes'.

Line 149: Change 'embedded' to 'taken up' or similar.

Line 167: to "gametocytes' sensitivity" or to "gametocyte sensitivity"

Line 175: change to 'activating the cAMP pathway '

Line 184: Fig ref should be to 4j. The authors could consider modelling whether this effect is statistically significantly greater than an additive effect (eyeballing it this seems likely), though these methods can be contentious.

Modelling whether this effect is significantly greater than an additive effect should be done by performing an isobologram analysis, however such analysis could be only applied with drugs that kill parasites at high concentrations, which is not the case of tadalafil

Line 192: 'interfering' can be read as downregulating NPPs, when the authors mean the opposite to facilitate drug uptake

Line 207: change to 'In support of the existence of NPPs'

Line 211: Do the authors mean invasion not egress?

Line 226: delete 'the' in 'clearance from the spleen'.

Line 229: It would be helpful if the authors could comment on whether the concentrations of tadalafil they have identified as relevant are likely to be achievable in patient serum. My back-of-the-envelope concentrations suggest they may not (but may well be wrong). Even if they are not this point can be raised, but the caveat should be noted.

In the revised manuscript we included the following sentence: *The uptake of Fluo-DHA was significantly increased with 10 μ M tadalafil, which approximately corresponds to ten-fold the reported peak serum concentration reached in humans after 60 min following 20 mg oral dose (C_{max} 378 ng/ml) (Smith 2013, PMID: 23869678).*

Line 257: spun - > centrifuged

Line 301: do the authors mean Fluo-DHA positive not GFP-positive?

Line 303. 2x10 not 2.10

Reviewer #2 (Remarks to the Author):

The manuscript by Bouyer et al analyses the ability of early and late sexual stage gametocytes to be permeable to isotonic solutes and Fluo-DHA when NPP blockers or drugs that regulate cAMP and hence activity of PKA are added to the culture medium. The authors propose that early stage gametocytes have functional new permeation pathways because of low PDE expression but that the nutrient channel is inhibited in mature gametocytes when PDE levels are high as levels of PKA phosphorylation, required for NPP activation, is low.

There are several concerns I have with the manuscript, mostly centred around the use of controls and the high concentrations of drugs that are used throughout. Consequently, it is difficult to assess whether the results they are seeing are truly meaningful and one can't be sure if the effects are a direct or indirect effect on NPP function. Whilst the RhopH products are transcribed in committed schizonts, it is important to demonstrate whether the RHopH proteins are actually expressed on early gametocytes. Are the authors proposing the NPPs differ between asexual stage and early sexual stages because the difference in sorbitol sensitivity is very large and moreover, artemisinin is not taken up by the NPP in the asexual stages.

In the revised supplemental Figure 1, we included an immunostaining experiment showing that the protein RhopH2 is expressed in immature GIE (Fig. S1d). This result is consistent with proteomics data shown on the PlasmoDB database reporting the detection of RhopH2 peptides by mass spectrometry in gametocytes.

Furthermore, to our knowledge there is no evidence in the literature that artemisinin is not taken up by NPPs in the asexual stages. Contrariwise, we have preliminary data showing that Fluo-DHA uptake

by trophozoites is significantly inhibited by the NPPs inhibitor NPPB, suggesting that NPPs contribute to Fluo-DHA uptake by trophozoites. We believe that this important result deserves further investigations that will be the focus of another manuscript, however these data are available for the reviewer if he requests them.

Revised Supplemental Figure S1

More specific comments are below:

Figure 1. Unlike trophozoite-stage parasites which rapidly lyse after 15 mins incubation in sorbitol, gametocyte sorbitol lysis was only double that of late rings. For this assay, how can the authors be confident that the lysis they are observing is not due to a small number of contaminating asexual stages?

We are confident that the observed lysis in Figure 1 is specific to stage II gametocytes because the read-out of sorbitol lysis experiments is performed by counting parasites before and after lysis on Giemsa-stained smears, and stage II gametocytes are easy to morphologically distinguish from trophozoites. In the revised manuscript we have included pictures of each gametocyte stage (Supplemental Figure S2) to illustrate this point.

a

	stage I	stage II	stage III	stage IV	stage V
average gametocytemia (%)	2.10	1.58	1.39	0.78	1.64
range (%)	1.19 - 3.45	0.55 - 4.31	0.58 - 2.07	0.5 - 1.3	0.68 - 3.18

b**c**
Revised Supplemental Figure S2

Why was the NPP assay assessed via Giemsa staining of parasites, rather than measuring RBC lysis in a spectrometer?

The NPP activity was assessed via Giemsa staining mainly because hemoglobin content within gametocyte-infected erythrocytes is strongly declining along gametocytogenesis (Hanssen et al 2012, PMID: 21945653), so hemoglobin release (and resulting absorbance at 540nm) would not have been a reliable method. Also gametocyte-infected erythrocytes were not purified before isosmotic lysis, and since lysis does not reach a plateau it would have been hard to set the 100% gametocyte-infected erythrocyte lysis. Moreover, as mentioned above the Giemsa staining analysis method allows to discriminate gametocytes from contaminating asexual stages and to measure the lysis for each specific gametocyte stage.

I was surprised the authors didn't use the GFP line for these experiments to determine what the percentage of GIE were in each sample.

The GFP line was only used to discriminate stage I gametocytes from trophozoites because this transgenic line was generated after we obtained results for other stages.

In part (a), where n=3, is this technical or biological repeats?

They are biological repeats. This is now specified in the legend of the revised manuscript.

Are the small changes in membrane conductance at -100mM in Fig (d) meaningful. How does this compare to ring-stage parasites for example, which don't have NPP? A comparison to ring stages would have been a very useful control.

We believe that the membrane conductance at -100 mV in gametocytes is meaningful because: 1/ it is significantly higher than uninfected erythrocytes, 2/ it is inhibited by NPPB, and 3/ it is declining between stage 1 and stage 5 (ANOVA test for trend $p=0.0013$, slope -134.7 ± 40.03 pS/stage transition).

We agree with the reviewer that it would have been interesting to compare with ring stages, however ring stages are very difficult to analyse by patch-clamp since they are hard to distinguish from uninfected cells without staining (chemical or genetic). Besides we would like to point out that NPPs are not absent in ring stages since NPPs activity in ring stages has been detected by sorbitol lysis experiments reported in the literature (Krugliak and Ginsburg, 2006, PMID: 16707126) as well as in our present study as early as 4 hours post-infection (Figure 1a).

For Fig 1f: It was not clear how viability was measured via luciferase (?measure release of luciferase into the culture medium).

As described in the material and methods section, cell viability was evaluated by adding a non lysing formulation of D-Luciferin substrate to transgenic gametocytes that express a luciferase enzyme, then luciferase activity was measured by quantifying the emitted luminescence (luciferase is able to convert luciferin into oxyluciferin via a chemical reaction, and some of the energy released by this reaction is in the form of light). This method has been validated to quantitatively assess the viability of gametocytes (Siciliano et al 2017, PMID: 28118506).

This may also be caused by parasites being more fragile. If viability was affected, washout experiments would show fewer transition to later stage gametocytes, was this observed?

As suggested, washout experiments were performed to show fewer transition to later stage gametocytes and these data are now included in the supplemental Figure S1.

Figure 2: Again, rings and trophs would have served as very useful controls on the graphs. None of the figures show uninfected erythrocytes (see line 108) but rings serve as better negative control.

Same answer as above, results with uninfected erythrocytes (obtained by both isosmotic lysis and patch-clamp experiments) have been included in revised Figures 1a and 1d, and we believe that rings are not proper negative controls.

Figure 3- the amount of H89 and 8-Br-cAMP used in this experiment seems very high at 100 micromM.

What concentrations have these PKA inhibitors have been used in other studies? It would have been good to do some titrations of these compounds and also to use asexuals as controls in these experiments.

The same amount of 8-Br-cAMP was used in our previous study (Ramdani et al 2015, PMID: 25951195), and previous work used H89 at 80 μ M or 100 μ M (Syin et al, 2001, PMID: 11559352; Sudo et al, 2008, PMID: 18501980).

Figure 4. The text on Ln 123-4 mentions revertant parasite line that has shed the overexpressing episome - was this shown as removing the drug doesn't necessarily remove episomes which can be very stable in parasites?

The same revertant parasite line was used in our previous study (Ramdani et al 2015, PMID: 25951195) where we showed the basal level of expression of PfPKAR in the revertant line.

What happens when the PDE inhibitors sildenafil or tadalafil is used on Stage 1 and II gams or an asexual stage parasites? If cAMP levels are high in the early gams then then the PDE inhibitors would not further increase isosmotic lysis. It would have been good to see this as a control.

As suggested, we have tested the effect of tadalafil on isosmotic lysis and FluoDHA uptake in stage II gametocytes. These results are now included in revised Figures S4 and S7.

Revised Supplemental Figure S4

Revised Supplemental Figure S7

DHA does not get taken up by the NPP in troph stages, so how does one envisage that the NPP may be used in early gametocytes but not in the asexual stages? It would have been good if trophs were used in this experiment as a negative control for these experiments.

To our knowledge there is no evidence in the literature that artemisinin is not taken up by NPPs in the asexual stages, and nowhere in the manuscript have we claimed this assertion. Contrariwise, we have preliminary data showing that Fluo-DHA uptake by trophozoites is significantly inhibited by the NPPs inhibitor NPPB, suggesting that NPPs contribute to Fluo-DHA uptake by trophozoites. We believe that this important result deserves further investigations that will be the focus of another manuscript, however these data are available for the reviewer if he requests them. For these reasons, we do not believe that trophozoites can be used as negative controls.

In gametocyte-infected erythrocytes, NPPs are part of the route for DHA uptake (as the fraction NPPB-susceptible), which also includes diffusion through bilayer or other route. The DHA permeability at the trophozoite stage is also probably an addition of several pathways, including NPPs that may represent only a minority of the pathway.

The amount of artemisinin used in the assay is very high, did you try any titration experiments? Likewise, there is no effect with Tadalafil at the 10 micromolar concentration.

Thanks for the suggestion, in the revised manuscript we used lower concentrations of artemisinin to measure the effect of NPPB on viability of stages II (revised Figure 4e): we used 700 nM that is the standard concentration used in Ring stage Survival Assays (PMID: 23208708). Since the IC₅₀ of artemisinin is higher in mature stages than in early stages, we kept the 5 μ M concentration for gamete egress experiment.

We performed new experiments to measure Fluo-DHA uptake upon tadalafil treatment, and as suggested by the reviewer 1 we re-analyzed all the uptake experiments to calculate the mean fluorescence intensity (MFI) values. These data now show a significant effect of Tadalafil at the 10 μ M concentration (revised Figure 4i).

The cells incubated with Fluo-DHA were only washed once prior to fixation so this may explain why the %Fluo-DHA positivity is so high. Again, how does a ring or trophozoite-stage parasite compare?

The % of Fluo-DHA positivity cannot be due to a lack of washing the cells since uninfected erythrocytes are totally negative for Fluo-DHA signal (Figure 4b). As mentioned above, asexual stages are not proper negative controls since there is no evidence about the role of NPP in DHA uptake in asexual stages.

Supp Fig 4 – How does the IC₅₀ of Fluo-DHA change going from early gametocytes to late gametocytes?

In the revised manuscript we show that IC₅₀ switches from 216 nM to 2.68 μ M between early and late gametocytes (revised supplemental Figure S5).

Revised Supplemental Figure S5

Reviewer #3 (Remarks to the Author):

In their manuscript “Plasmodium falciparum sexual parasites regulate infected erythrocyte permeability” Bouyer et al. investigate the presence of NPPs in Plasmodium falciparum gametocytes. They use a variety of techniques, including patch clamping, imaging and survival assays to determine if the NNPs are active in the sexual stage of the parasite. Furthermore, they investigate whether artemisinin requires the NPP to enter the infected erythrocyte. In their final experiments, the authors use clever genetic experiments coupled with the use of inhibitors and activators to show that the activity of the NNP requires the activity of the parasite kinase PKA. The authors conclude with a model about the regulation of the NPP by PKA. The authors represent the data clearly (the color coding throughout makes the figures easy to follow) and provide good controls throughout. There are several novel aspects to these findings, but not all of the experiments are completely convincing. There are several additional experiments that the authors might consider to strengthen their findings.

-The authors describe how the gametocytes culture is established and purified, but do not show any images of the resulting gametocyte cultures. From the materials and methods section, it appears that the authors do not induce gametocytogenesis through metabolic depletion with spent medium or lysophosphatidylcholine depletion, but instead solely rely on the removal of asexual parasites using extended NAG treatment. Hence, the starting parasitemia of gametocytes is likely to be very low. As

NAG treatment does not remove asexual stage parasites within one cycle (Miao et al. <http://dx.doi.org/10.1016/j.exppara.2013.09.010>), there are likely to be asexual stage parasites in the gametocyte culture; with a low parasitemia of gametocytes, even if a low number of asexual parasites survive, these will likely still form a relatively large fraction of the population. Without knowing how pure the gametocyte cultures are, the lysis of parasites shown in Figure 1 and 2 could be explained by the presence of trophozoites or schizonts in the culture. As the level of asexual parasites will drop over time in the presence of NAG, this could explain the drop in the lysis in the culture over time. The authors should give an indication of the percentage of gametocytes in their culture that is achieved at each stage with their method, either by flow cytometry using the strain they described in which GFP is controlled by the pfs16 promoter or several large fields of purified parasites stained either with Giemsa or anti-Pfs16 (or similar gametocyte-specific antibody).

Our protocol to produce gametocytes is based on overgrowing the parasite culture and is the most commonly used method to obtain large numbers of gametocytes (Delves et al, PMID: 27560172; Fivelman et al, PMID: 17521751). Using this method and the NF54 strain, we routinely obtain gametocytemia between 1 and 5%. We agree with the reviewer that some contaminating asexual parasites may persist in the cultures for a couple of days, however this is not an issue because gametocytes from stage II to V are very easy to morphologically distinguish from asexual stages. Our method to quantify lysis is based on counting gametocytes on Giemsa-stained smears, and in the electrophysiology experiments we choose the cells to patch according to their morphology. The only possible confusion could be between stage I gametocytes and trophozoites; for this reason we quantified lysis and membrane conductance on stage I from the transgenic line that selectively expresses GFP in gametocytes.

To clarify these points, in the revised manuscript we included in Supplemental Figure S2 several pictures to show the morphologic differences of each gametocyte stages and a table showing the gametocytemia at each stage for the cultures used in the isosmotic lysis experiments.

-Throughout, the authors rely on a relatively indirect measure (lysis of parasites) to investigate the presence of the NPP. In the introduction, they mention the role of the high molecular weight rhoptry complex in the induction of the NPP. A much more direct way to look at the presence of the NPP would be immunofluorescence microscopy using antibodies against members of this complex. The authors reference two papers that show that the high molecular weight rhoptry complex is present in schizonts in the introduction, but both papers describe asexual parasites; there is (to my knowledge) no information about the presence of this complex in schizonts that are committed to form gametocytes.

Isosmotic lysis and patch clamp experiments are the standard methods to investigate the presence of NPPs in Plasmodium-infected erythrocytes and are widely accepted in the field (Staines et al Int J Parasitol 2007, PMID: 17292372). To strengthen our data, as suggested by the reviewer, we performed an immunofluorescence microscopy experiment showing that the protein RhopH2 is expressed in immature GIE (revised Fig. S1D). This result is consistent with proteomics data shown on the PlasmoDB database reporting the detection of RhopH2 peptides by mass spectrometry in gametocytes.

-The authors could also use PKG inhibitors to remove asexual parasites, which may act faster than NAG (Portugaliza HP et al. Reporter lines based on the gexp02 promoter enable early quantification of sexual conversion rates in the malaria parasite Plasmodium falciparum. doi: 10.1038/s41598-019-50768-y).

Thanks for the suggestion, we will test this method in our lab to obtain the future gametocytes.

-Related to this, the authors use fluorescent derivatives of artemisinin to show uptake of compounds, but this is a rather non-standard assay and, as the authors acknowledge, if artemisinin uses the NPP, it is not the only pathway for the drug to enter the cells. A more common way to show uptake by the NPP is through the use of PPIX (Sigala PA, Crowley JR, Henderson JP, Goldberg DE. 2015. Deconvoluting heme biosynthesis to target blood-stage malaria parasites. *eLife* 4:e09143. doi: 10.7554/eLife.09143). This would allow for costaining with gametocyte markers or the visualization of GFP produced using a gametocyte-specific promoter. In conjunction with the NPP inhibitors that the authors use, the use of PPIX would show definitively that the NPP is active in gametocytes.

We used fluorescent derivatives of artemisinin to investigate the role of NPP in the uptake of antimalarials, but not to demonstrate the existence of NPPs. As mentioned above, isosmotic lysis and patch clamp experiments are the standard methods to investigate the presence of NPPs in *Plasmodium*-infected erythrocytes and are widely accepted in the field, whereas the use of PPIX is more controversial.

-The results in Figure 1D are hard to interpret without data on uninfected cells. What is the level of conductance in uninfected cells?

In the revised manuscript we included patch-clamp experiments showing membrane conductance on uninfected erythrocytes (revised Figure 1d).

-The authors should mention that asexual ring-stage parasites, in which the NPP is not yet active, are sensitive to artemisinin. Hence, artemisinin can penetrate the erythrocyte without NPP and kill parasites.

The assumption that NPPs are not active at ring stage should require more investigations. Indeed, Krugliak and Ginsburg (PMID: 16707126) showed the starting of NPP activity via isosmotic lysis at ring stages. Our own results (Figure 1a) shows that early rings and late rings display a permeability to sorbitol much higher than uninfected cells – also suggesting a very early start for NPPs activity during asexual cycle, but not at full rate.

-The PfPDE δ line that the authors use is a 3D7 derivative and hence may convert to gametocytes at a vastly different rate than the NF54 strain used by the authors. This caveat should be mentioned. With the advent of genetic manipulation of malaria parasites, a complemented PfPDE δ line could be made that would provide a better comparison (admittedly still a lot of work).

Data presented in this manuscript do not focus on sexual conversion but rather on existence of NPPs. The PfPDE δ line was only used in isosmotic lysis experiments in which the read out is the % of lysis calculated as $[1 - (\text{gametocytemia (t60)} / \text{gametocytemia (t0)})] * 100$. Thus, any difference in gametocyte production between NF54 and the PfPDE δ line does not influence the result.

-The model that the authors propose in Figure 5 would benefit from additional details. The white arrow before 'active PKA' actually passes two membranes. As neither PKA nor cAMP can pass membranes and PKA is not thought to be exported, it is not clear how this step would occur. Other than the export of proteins and the formation of Maurer's clefts, no pathway for modification or signaling to the host cell has been described. The authors should provide an explanation how they think this step takes place.

We agree that it is not totally clear how this step would occur, however we would like to point out that according to Desai et al (PMID: 9050902), the pore size of the malaria parasite's nutrient channel allows passage of soluble macromolecules of up to 1400 Da, therefore cAMP could be exported from parasite to the erythrocyte. Regarding PKA, there is still a debate whether the PKA activity described in our manuscript, as well as in our previous work (Merckx et al, 2008_PMIID: 18248092; Ramdani et al, 2015 PMID: 25951195; Naissant et al, 2016, PMID: 27136945), is of parasitic or human origin. In either case, our conclusion that "cAMP signaling regulates NPPs activity" is still valid. To clarify this point, we included in the revised manuscript "parasite or human PKA" in the legend of Figure 5 and we modified the color of the arrow to avoid the misunderstanding that PKA is necessarily exported by the parasite.

REVIEWERS' COMMENTS:

Reviewer #1 (Remarks to the Author):

(For the record, I was unable to open the Reporting Summary file or the Editorial policy checklist)

My concerns have been addressed, and I think this is an important and rigorous study. I have some remaining small suggestions for changes to wording for the authors to consider.

1.

"Synchronous production of specific 329 gametocytes stages was achieved by treating synchronized cultures at the ring stage (10-15% 330 parasitemia, day 0) with 50 mM N-acetylglucosamine (NAG) for 5 days to eliminate asexual parasites. 331 Gametocyte preparations were enriched in different experiments by magnetic isolation. Stage I GIE 332 were collected at day 1 post NAG treatment, stage II GIE were collected at days 2 and 3, stage III GIE 16 333 were collected at days 4 and 5, stage IV GIE were collected at days 6 and 7, and stage V GIE were 334 collected at day 8 onwards."

In this context "day 1 post NAG treatment" implies to the reader that the 5 days have elapsed. Please change to "day 1 after initiating NAG treatment" or similar.

I actually asked about stage II GIEs in my previous review, raising a similar concern to other reviewers on the possibility contaminating asexual parasites. Given the commonality of this concern the authors might consider adding a comment in the methods to highlight that they morphologically confirmed that they were counting gametocytes.

2.

I thank the authors for paying such attention to my "change NPP to NPPs suggestion", unfortunately I fear I have caused confusion. I would suggest reverting to "NPP" in the cases where NPP is used as a modifier for another word immediately after, e.g. "NPP activity", "NPP components". I apologise to the authors for the lack of clarity in the first suggestion.

3.

Fig S6: the use of a comma as a decimal point may be confusing to an international readership

Reviewer #2 (Remarks to the Author):

The authors have taken the reviewers suggestions on board when revising the manuscript. With respect to the measuring %lysis of GIE, the authors have done this via Giemsa staining, which they explained why they did it this way. Given that the counts of gams are very low before the addition of compound, it is important to indicate how many GIE were counted in the analysis in the methodology.

Fig 3: Ln 152 states that the PKA inhibitors compounds strongly ablate GIE permeability when in fact there is only a 2-3 fold reduction of % lysis. This dramatic reduction occurs when NPPB is added. This sentence needs to be revised. This suggests that PKA activity, while having some contribution, is not essential for NPP activity.

Reviewer #3 (Remarks to the Author):

In their revised manuscript, the authors go to admirable lengths to address the queries to the first version of the manuscript – the localization of RhopH2 in the GIE is particularly convincing. Many of the initial comments have been addressed. However, there are some remaining questions that could be clarified.

-The reference that the authors provide to state that NPP activity may be present in ring-stage parasites (Krugliak and Ginsburg (PMID: 16707126)) does not support this conclusion as firmly as might be expected. In that paper, the authors state in the discussion: "In all cases and with both experimental protocols, only a small fraction of the young developmental stages do lyse. We interpret this result to mean that not all infected cells develop NPPs at the same time. With time in the parasite life cycle, NPPs gradually appear in most (and sometimes in all) infected cells, as evidenced by the fact that the fraction of cells that lyse increases with parasite maturation (Fig. 3)." As all ring stage parasites are sensitive to artemisinin, the NPPs are very unlikely to be an important portal for entry of artemisinin into the parasite. It would be useful to point this out in the text.

-The modification of the model goes some way to resolving the issue of the transport of the cAMP to the host cell, but not fully. Whereas the authors are correct that the PVM allows the passive transport of molecules with MW <1200 Da, the parasite plasma membrane is impermeable to cAMP. Even if it is host PKA that is activated by parasite-derived cAMP, it remains unclear how a molecule generated in the parasite can have an effect at the erythrocyte surface. It would be useful if the authors could expand on their model to indicate how they believe the increased levels of cAMP are causing an effect in the host cell.

-The authors state in the manuscript that hemoglobin metabolism reduces as the gametocytes mature. As hemoglobin breakdown is required for the activation of artemisinin (and likely the retention of Fluo-DHA), can the authors rule out that the decrease of activity and uptake of the drug reflects this decrease, rather than a decrease in NPP activity? The decrease in activation would also explain why the fluorescent compound is detected at lower levels in the more mature gametocytes. In addition, can the authors rule out that the pretreatment with NPP inhibitors does not slow down hemoglobin breakdown, leading to lower levels of retention of Fluo-DHA in the parasite?

REVIEWERS' COMMENTS:

Reviewer #1 (Remarks to the Author):

My concerns have been addressed, and I think this is an important and rigorous study.

We thank the reviewer for his/her positive comment.

I have some remaining small suggestions for changes to wording for the authors to consider.

1. "Synchronous production of specific 329 gametocytes stages was achieved by treating synchronized cultures at the ring stage (10-15% 330 parasitemia, day 0) with 50 mM N-acetylglucosamine (NAG) for 5 days to eliminate asexual parasites. 331 Gametocyte preparations were enriched in different experiments by magnetic isolation. Stage I GIE 332 were collected at day 1 post NAG treatment, stage II GIE were collected at days 2 and 3, stage III GIE 16 333 were collected at days 4 and 5, stage IV GIE were collected at days 6 and 7, and stage V GIE were 334 collected at day 8 onwards." In this context "day 1 post NAG treatment" implies to the reader that the 5 days have elapsed. Please change to "day 1 after initiating NAG treatment" or similar.

Thanks, this has been modified in the revised manuscript.

I actually asked about stage II GIEs in my previous review, raising a similar concern to other reviewers on the possibility contaminating asexual parasites. Given the commonality of this concern the authors might consider adding a comment in the methods to highlight that they morphologically confirmed that they were counting gametocytes.

In the revised manuscript, we included the following sentence in line 351: "Microscopy examination of Giemsa-stained smears allowed to morphologically confirm that only gametocytes, but not contaminating asexual parasites, were counted."

2. I thank the authors for paying such attention to my "change NPP to NPPs suggestion", unfortunately I fear I have caused confusion. I would suggest reverting to "NPP" in the cases where NPP is used as a modifier for another word immediately after, e.g. "NPP activity", "NPP components". I apologise to the authors for the lack of clarity in the first suggestion.

Thanks, this has been modified in the revised manuscript.

3. Fig S6: the use of a comma as a decimal point may be confusing to an international readership

Thanks, this has been modified in the revised manuscript.

Reviewer #2 (Remarks to the Author):

The authors have taken the reviewers suggestions on board when revising the manuscript. With respect to the measuring %lysis of GIE, the authors have done this via Giemsa staining, which they explained why they did it this way. Given that the counts of gams are very low before the addition of compound, it is important to indicate how many GIE were counted in the analysis in the methodology.

This is mentioned in line 349: "Parasitemia was estimated for each point by counting infected cells out of at least 4000 erythrocytes."

Fig 3: Ln 152 states that the PKA inhibitors compounds strongly ablate GIE permeability when in fact there is only a 2-3 fold reduction of % lysis. This dramatic reduction occurs when NPPB is added. This sentence needs to be revised. This suggests that PKA activity, while having some contribution, is not essential for NPP activity.

We removed the word "strongly" from this sentence.

Reviewer #3 (Remarks to the Author):

In their revised manuscript, the authors go to admirable lengths to address the queries to the first version of the manuscript – the localization of RhopH2 in the GIE is particularly convincing. Many of the initial comments have been addressed. However, there are some remaining questions that could be clarified.

We thank the reviewer for his/her positive comment.

-The reference that the authors provide to state that NPP activity may be present in ring-stage parasites (Krugliak and Ginsburg (PMID: 16707126)) does not support this conclusion as firmly as might be expected. In that paper, the authors state in the discussion: “In all cases and with both experimental protocols, only a small fraction of the young developmental stages do lyse. We interpret this result to mean that not all infected cells develop NPPs at the same time. With time in the parasite life cycle, NPPs gradually appear in most (and sometimes in all) infected cells, as evidenced by the fact that the fraction of cells that lyse increases with parasite maturation (Fig. 3).” As all ring stage parasites are sensitive to artemisinin, the NPPs are very unlikely to be an important portal for entry of artemisinin into the parasite. It would be useful to point this out in the text.

We agree with the reviewer that data from Krugliak and Ginsburg suggest that a proportion of young infected cells are devoid of NPPs, however they do detect NPPs in some ring-stages; and we believe that our present data also show the existence of NPPs in a proportion of ring stages. Nevertheless, we would like to point out that nowhere in the manuscript we claim that NPPs are involved in artemisinin uptake by asexual stages. Although characterizing the role of NPPs in drug uptake by asexual stages would be very interesting, we believe that it is out of the scope of this study which is focused on sexual stages. To clarify this point, we modified the sentence line 220 : “...our results provide evidence that NPPs significantly contribute to the uptake of artemisinin derivatives in GIE.”

-The modification of the model goes some way to resolving the issue of the transport of the cAMP to the host cell, but not fully. Whereas the authors are correct that the PVM allows the passive transport of molecules with MW <1200 Da, the parasite plasma membrane is impermeable to cAMP. Even if it is host PKA that is activated by parasite-derived cAMP, it remains unclear how a molecule generated in the parasite can have an effect at the erythrocyte surface. It would be useful if the authors could expand on their model to indicate how they believe the increased levels of cAMP are causing an effect in the host cell.

According to Wentzinger et al, (PMID : 18590734), “TMHMM predictions for two of the PDEs, PfpPDE β and PfpPDE δ , indicate that their catalytic domains might be exposed outwards of the parasite, i.e. towards the lumen of the parasitophorous vacuole. The same type of analysis predicts that the catalytic domains of PfpPDE α A, PfpPDE α B and PfpPDE γ are intracellular, i.e. exposed towards the cytoplasm of the parasite. If confirmed, this topology of the four PDEs might provide the opportunity for the parasite to not only regulate its own cyclic nucleotide signaling network, but also to interfere with that of its environment.”

Therefore, the plasmodial PfpPDE δ may regulate the levels of cAMP in the host erythrocyte that in turn may activate the host PKA. We thank the reviewer to raise this point and we modified our model accordingly, lines 273-276 and Figure 5.

-The authors state in the manuscript that hemoglobin metabolism reduces as the gametocytes mature. As hemoglobin breakdown is required for the activation of artemisinin (and likely the retention of Fluor-DHA), can the authors rule out that the decrease of activity and uptake of the drug reflects this decrease, rather than a decrease in NPP activity? The decrease in activation would also explain why the fluorescent compound is detected at lower levels in the more mature gametocytes. In addition,

can the authors rule out that the pretreatment with NPP inhibitors does not slow down hemoglobin breakdown, leading to lower levels of retention of Fluo-DHA in the parasite?

For the first point we agree with the reviewer and we included the following sentence in the revised manuscript, line 217: *“(4, 28). Although we cannot rule out that the decrease in Fluo-DHA activity in mature stages reflects the slowdown of hemoglobin breakdown required for the activation of artemisinin (Klonis et al, PNAS 2011), our results provide evidence that NPPs significantly contribute to the uptake of artemisinin derivatives in GIE.”*

However, we believe that the pretreatment with NPP inhibitors is very unlikely to slow down hemoglobin breakdown, first because NPPB and furosemide are highly specific inhibitors of transporters therefore it is unlikely that they interfere with the enzymatic process of hemoglobin digestion, and second because the short preincubation time with NPP inhibitors would not allow to slow down the hemoglobin digestion process which takes several days.